

# DeepMethylation: a deep learning based framework with GloVe and Transformer encoder for DNA methylation prediction

Zhe Wang[1,*], Sen Xiang[1,*], Chao Zhou[2] and Qing Xu[2]

[1] Wuhan University of Science and Technology, Wuhan, Hubei, China
[2] China Three Gorges University, Yichang, Hubei, China
* These authors contributed equally to this work.

## ABSTRACT

DNA methylation is a crucial topic in bioinformatics research. Traditional wet experiments are usually time-consuming and expensive. In contrast, machine learning offers an efficient and novel approach. In this study, we propose DeepMethylation, a novel methylation predictor with deep learning. Specifically, the DNA sequence is encoded with word embedding and GloVe in the first step. After that, dilated convolution and Transformer encoder are utilized to extract the features. Finally, full connection and softmax operators are applied to predict the methylation sites. The proposed model achieves an accuracy of 97.8% on the 5mC dataset, which outperforms state-of-the-art methods. Furthermore, our predictor exhibits good generalization ability as it achieves an accuracy of 95.8% on the m1A dataset. To ease access for other researchers, our code is publicly available at https://github.com/sb111169/tf-5mc.

## INTRODUCTION

Epigenetics is first proposed to investigate the heritable changes in the regulation of gene expression without altering the nucleotide sequence of DNA. Researchers have discovered various epigenetic mechanisms, including protein acetylation and methylation (*Zhang, Lu & Chang, 2020a*). Currently, N6-methyladenine (6mA), N4-methylcytosine (4mC), and 5-methylcytosine (5mC) are the three most widely studied types of DNA methylation. Take 5mC as an example, it commonly appears on the fifth carbon atom of cytosine in the DNA sequence's CpG dinucleotides. DNA methyltransferase transfers the methyl (-CH3) group from S-AdenosylMethionine (SAM) to the fifth carbon atom of cytosine (*Adampourezare et al., 2021*).

Studies have indicated the possible negative impact on organisms of abnormal DNA methylation. Firstly, DNA methylation affects the level of gene expression and even leads to gene silencing or abnormal expression (*Ehrlich, 2003*). For example, DNA methylation can change the conformation of chromatin, thus affecting chromatin accessibility and gene expression. In addition, the risk of gene mutations is positively correlated with DNA methylation (*De Bont & Van Larebeke, 2004*). Methylation sites are prone to be damaged

Corresponding authors
Sen Xiang, xiangsen@wust.edu.cn
Chao Zhou, zhouchao@ctgu.edu.cn

in the process of replication and repair of DNA. If they are not repaired correctly, it may lead to loss of DNA or accumulation of mutations. Moreover, the same is true of the occurrence and development of cancer (*Xu et al., 2011*; *Chowdhury et al., 2011*; *Lu et al., 2012*; *Koivunen et al., 2012*). Some cancer cells have aberrant methylation of genes involved in important cellular life processes such as cell growth, differentiation and apoptosis, suggesting that DNA methylation may promote tumor initiation and progression. For instance, mutations in IDH1/2 produce the oncogenic metabolite 2-HG, which results in increased DNA methylation at the cellular level. This alteration affects gene expression and leads to cancer. Finally, embryonic development and adult diseases are also associated with DNA methylation (*Jin et al., 2008*; *Tatton-Brown et al., 2014*; *Baets et al., 2015*). DNA methylation plays an important role in embryonic development, and abnormal methylation may cause birth defects or abnormal development. The status of three functional protein families in the epigenetic system (write, reader, eraser), and their associated genes' genetic variation can cause diseases (*e.g.*, autism, blood disease) by affecting overall cell-level epigenetics. Therefore, DNA methylation plays an important role in gene expression regulation and chromatin structure variation, and the detection of methylation is of great importance.

Current methods for methylation detection include wet experiments, traditional machine learning methods, and deep learning methods. Wet experiments conduct molecular biology tests to distinguish between methylation and demethylation in DNA samples. This typically involves bisulfite treatment (*Smallwood et al., 2014*; *Kernaleguen et al., 2018*), enzymatic digestion, and chromatin immunoprecipitation. Following bisulfite treatment, methylated cytosine is oxidized and transformed to unmethylated uracil, whereas unmethylated cytosine remains unchanged, and the difference indicates methylation.

Traditional machine learning methods generally consist of three steps: data processing, feature extraction, and classification, which are all designed based on the experience of the researchers. Commonly-used features include physical, statistical, and sequence annotation features such as base frequencies, G+C content, length, repetitive sequences, RNA elements, and protein binding sites (*Fang et al., 2006*; *Zhang et al., 2015*). Based on the features, classification algorithms such as logistic regression, support vector machine, and decision trees are used to identify the methylation sites.

In contrast, deep learning methods are more straightforward. Instead of manually specifying the feature extractor and classifier, deep neural networks can automatically extract features and predict methylation results from DNA sequences. Furthermore, driven by datasets with a large number of samples, deep learning can extract more essential features than manually designed models. For instance, the DNA module and the CpG module of the DeepCpG model (*Angermueller et al., 2017*) can predict the relationship between DNA sequences and their methylation status, as well as the relationship between adjacent CpG sites within a single cell or across cells.

In general, wet experiments achieve high accuracy, but they only predict a small number of DNA methylation sites. In addition, conducting wet experiments requires not only great cost and time, but also professional knowledge in biology, and these factors make it
difficult to be widely applied. Traditional machine learning needs specified feature design, which also requires professional experience and extensive tests to find good feature descriptors. In contrast, deep learning methods can automatically learn the most relevant features without specifying them in advance, and can handle large datasets and high-dimensional data.

Although deep learning methods provide new insights to detect DNA methylation, they still face challenges. On the one hand, convolutional neural networks (CNNs) are not sensitive to 1D sequential data such as DNA. On the other hand, recurrent neural networks (RNNs) perform better in extracting sequential features, but they are not good at exploring relationships for bases far away. In addition, the current DNA encoding methods, one-hot and word embedding, emphasize local information and ignore global relationships.

To solve these problems, in this article, we propose DeepMethylation, a novel deep-learning based scheme to predict DNA methylation sites. The contribution of this article is as the following. Firstly, with word embedding and GloVe, we propose a novel DNA encoding method. This new representation format improves the ability in modeling the relationship between DNA sub-sequences. Secondly, dilated convolution and Transformer encoder are incorporated to better extract both local and global features, especially the relationship between DNA sequences far from each other. Last but not least, dense full connections are used to predict the methylation status of each site. Experimental results demonstrate that the accuracy of the proposed method reaches 97.8%, which outperforms other state-of-the-art methods.

## RELATED WORKS

### Wet experiments

Genome-wide single nucleotide resolution (GWGSR) typically requires wet experiments to be realized. Currently, the main approaches for achieving GWGSR include whole-genome bisulfite sequencing (WGBS), reduced representation bisulfite sequencing (RRBS), and DNA methylation chip.

WGBS (*Smallwood et al., 2014*; *Kernaleguen et al., 2018*) is a high-resolution and comprehensive method for full-genome sequencing *via* bisulfite treatment, which converts unmethylated cytosine to uracil, but does not convert methylated cytosine. Methylation status of individual cytosines can be determined at the single nucleotide level by comparing the DNA sequences with and without bisulfite treatment. RRBS (*Guo et al., 2013*; *Farlik et al., 2015*; *Hou et al., 2016*) is a cost-effective alternative to WGBS and involves sequencing the CpG-rich subset of the genome. RRBS reduces the requirement for sequence depth to cover the entire genome and still provides single nucleotide resolution at CpG sites. DNA methylation chips (*Morris et al., 2014*) represent microarray-based platforms that simultaneously detect DNA methylation levels among thousands of CpG sites in the genome. These chips contain probes that are specific to methylated or unmethylated CpG sites, and the intensity of the signal from each probe indicates the site's methylation level.

Although wet experiments produce accurate prediction results, it needs great financial cost and time, as well as professional biology knowledge, which is inefficient in implementation.

## Traditional machine learning methods

With the rapid advancement in automatic DNA sequencing technology, huge amounts of DNA sequences are obtained, promoting the analysis of DNA data. Traditional machine learning methods involve two steps. Firstly, manually designed DNA features are proposed. After that, with these features, machine learning classification algorithms are utilized to predict the methylation. *Stevens et al. (2013)* integrated the features from chromatin immunoprecipitation sequencing and methylation-sensitive restriction enzyme sequencing, and predicted the methylation status of CpG sites in the human genome by using a conditional random field model. *Zhang et al. (2015)* utilized various features, including methylation markers, genomic locations, and regulatory factors, to design a methylation prediction model with a random forest classifier. *Fang et al. (2006)* developed a CpG island methylation prediction tool called MethCGI using CpG island data from the human brain. This model takes input features such as CpG ratio, GC content, TpG frequency, and transcription factor binding site distribution, and employs a support vector machine as a classifier.

Machine learning methods have demonstrated higher efficiency and lower costs than wet experiments. However, the performance of machine learning models is limited by the manual selection of feature descriptors and classifiers, which relies on the experience of the researchers.

## Deep learning methods

In recent years, with the rapid development of neural networks, deep learning methods have been applied to DNA methylation prediction (*Routhier & Mozziconacci, 2022*). Different from traditional machine learning that highly relies on the experience of the researchers, deep learning methods, including convolutional neural networks (CNN) and recurrent neural networks (RNN), can automatically learn the essential features from the raw sequential data, and construct end-to-end models, which have been proven to outperform traditional machine learning methods in predicting DNA methylation sites.

*Angermueller et al. (2017)* proposed the DeepCpG model. The model consists of DNA module, CpG module, and joint module. The DNA module involves two convolutional layers and a pooling layer to identify correlations between DNA sequence patterns and methylation status. The CpG module employs a bidirectional gated recurrent network to identify correlations between adjacent CpG sites. The joint module learns the interaction between the DNA and CpG modules to predict the methylation status in all cells. *Tian et al. (2019)* proposed MRCNN, which used the correlation between DNA sequence patterns and methylation levels to predict the methylation of the CpG site at single base resolution. The model used one-hot encoding, convolution, pooling, and fully connected layers to output the predicted value. With a continuous loss function, MRCNN achieves smooth regression of methylation value, and produces more accurate results than the

DeepCpG model. *Zhou (2020)* built an RNN-based DNA methylation prediction model, this model first converts the raw DNA sequence into matrix data through one-hot encoding, and then sends it to the RNN model for feature extraction and methylation prediction. The results have shown that RNNs are more suitable for handling sequence data and extracting hidden temporal features from the sequence than CNNs. *Cheng et al. (2021)* proposed iPromoter-5mC, believing that DNA chemical properties can affect its genetic traits. To address this issue, they combined one-hot encoding with deoxyribonucleic acid nucleotide properties and their frequency (DPF) to generate a composite feature set. They then used a deep neural network to process the composite feature set for identifying methylation modification sites in promoters. Tran et al. considered that the DNA sequence can be regarded as a distinct linguistic system, and they proposed an efficient encoding method to identify 5-methylcytosine sites. By embedding k-mers, they transformed the DNA sequence into 'sentences', and then generate the feature vector of the DNA sequence (*Nguyen et al., 2021*) with k-mers representation. Then, the feature vectors were separately sent to xgboost, random forest, deep forest and deep feedforward neural network. The final results showed that the performance of this model was better than iPromoter-5mC.

In general, deep neural networks have better learning abilities than traditional learning methods, and thus produce more accurate results. Nevertheless, CNNs and RNNs still encounter challenges in encoding feature representations and efficiently extracting global long-distance features, and further research is desired.

## MATERIALS AND METHODS

### The overall framework

As shown in Fig. 1, the proposed DeepMethylation has three modules, which are data processing module, feature extraction module and classification module. First, in data processing module, the one-dimensional DNA sequence is segmented and converted to a $39 \times 300$ matrix with word embedding and GloVe. After that, the feature extraction module utilizes Transformer encoder and dilated convolution to extract global and local features. Finally, with the extracted features, the classification module predicts the methylation state of each site of the DNA sequence.

### Data processing

As a long sequence, DNA is not conducive to presenting the relationship among different fragments. Therefore, the first step is to convert a one-dimensional DNA sequence to a group of short fragments. Although one-hot encoding (*Abbas, Tayara & Chong, 2021*) can represent each base of DNA as a binary bit, it cannot provide the sequence orders or measure the distance (*Huang et al., 2021*) between related words. In this article, word embedding and GloVe algorithm are used to better model the relationship in DNA sequences.

By following the rule of WGBS, the golden standard of methylation detection, DNA sequences are cropped into 41 bp segments. As shown in Fig. 2, a 3-bp window slides over

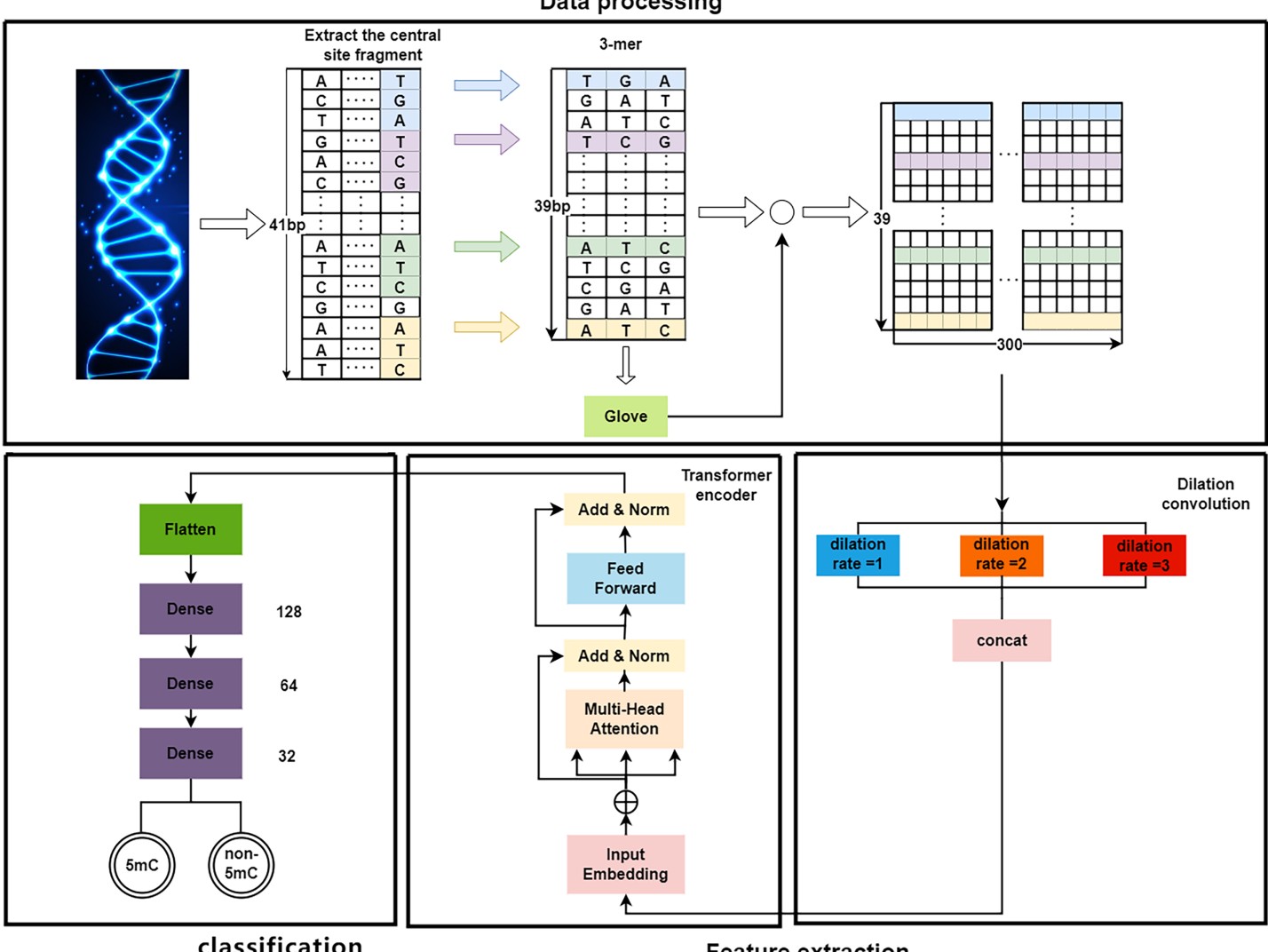

**Figure 1** The overall framework of DeepMethylation.

a segment and produces a series of 3-mer sub-sequences. As a result, a 41 bp DNA sequence is converted to a 39 × 3 3-mer matrix.

To explore the relationship between these 3-mer fragments, GloVe (*JeffreyPennington & Manning, 2014*; *Wang et al., 2022*), a word embedding model based on global vectors, is utilized. It first checks the context of neighboring 3-mer fragments and obtains a co-occurrence matrix. Figure 3 depicts an example of the co-occurrence matrix for three four-word sentences. Take the combination of 'CGG-ATC' as an example, it happens twice that 'CGG' appears before 'ATC', and the corresponding intersection with the row index 'CGG' and the column index 'ATC' is valued at two, which is marked in pink in the co-occurrence matrix. Mathematically, the co-occurrence matrix is notated as $X$, and $X_{i,j}$ represents the frequency of word $j$ appearing after $i$. In this way, the example in Fig. 3 can also be noted as $X_{CGG,ATC} = 2$. Moreover, it is noteworthy that the matrix is symmetric

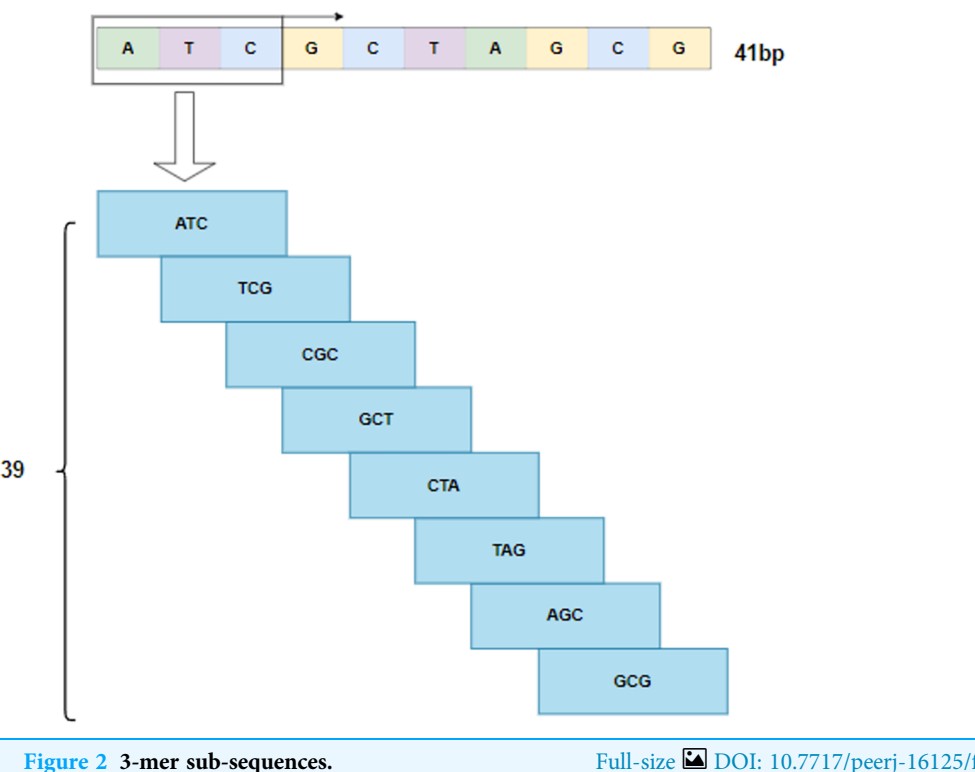

**Figure 2 3-mer sub-sequences.**               

about the diagonal line, and the elements in the upper-right of the matrix are computed and copied to the lower-left.

In Fig. 3, the co-occurrence matrix only indicates the relationship of three sub-sequences, and in order to model the relationship for all sub-sequences, GloVe algorithm traverses the entire *corpus* and derives a global word vector dictionary through inner product operation and translation transformation of words (*Cochez et al., 2017*; *Liu et al., 2019a*), which makes the mapping values equal or approximate to the co-occurrence probability of words. To be specific, an energy function $J$ is defined as

$$J = \sum_{i,j=1}^{N} f(X_{i,j})\left[V_i^T \widetilde{V}_j + b_i + b_j - log(X_{i,j})\right]^2 \tag{1}$$

where $b_i$ and $b_j$ are offsets, and $N$ is the total number of words. $V_i$ represents the word vector in the global dictionary to be obtained, $\widetilde{V}_j$ is the separate context vector that helps solve $V_i$. Since $J$ is a convex function, $V_i$ can be solved *via* optimization algorithms such as gradient descent. In addition, the weighting factor $f(X_{i,j})$ is defined as

$$f(X_{i,j}) = \begin{cases} \left[\frac{X_{i,j}}{T_X}\right]^{\alpha} & \text{if } X_{i,j} < T_X \\ 1 & otherwise \end{cases} \tag{2}$$

With the truncation parameter $T_X$ and the non-linear mapping parameter $\alpha$, the model can retain crucial information in the co-occurrence count while eliminate noise and irregular co-occurrence. In very special cases, $f(X_{i,j})$ equals to 1 only when two words are

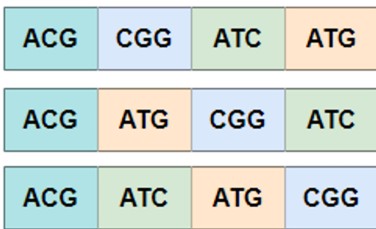

DNA 3-mer sub-sequences

| counts | ACG | ATG | CGG | ATC |
|--------|-----|-----|-----|-----|
| ACG | 0 | 1 | 1 | 1 |
| ATG | 1 | 0 | 2 | 2 |
| CGG | 1 | 2 | 0 | 2 |
| ATC | 1 | 2 | 2 | 0 |

**Figure 3 Co-occurrence matrix.**

semantically similar and locate closely to each together in the vector space. $\alpha$ is an empiric value that equals to 0.75, as has been proved can yield good perfromance (*JeffreyPennington & Manning, 2014*).

In the implementation, the length of $V_i$ is set to 300 for each valid vector word. Once all word vectors $V_i$ are obtained, each 3-mer word can be represented with the corresponding word vector. As a result, the 39 3-mer words in Fig. 1 can be presented with a $39 \times 300$ encoding vector matrix.

## Feature extraction

After data processing, the $39 \times 300$ word vector matrix is used for feature extraction. To be specific, this matrix is regarded as a word vector embedding layer that is utilized as input for the feature extraction module, which utilizes dilated convolution and Transformer encoder as shown in Fig. 1.

On the one hand, to enlarge the receptive field while keeping low computational complexity (*Liu et al., 2019b*; *Yuan et al., 2019*), dilated convolution is utilized. As shown in Fig. 4, in dilated convolution, the filter is expanded by inserting zeros between its values. This effectively increases the receptive field of the filter without increasing the number of parameters, allowing it to capture larger spatial structures and longer-term dependencies of the input. In this study, three branches with dilation rates of 1, 2, and 3 are used, followed by feature concatenation, producing features of contextual information at different scales.

On the other hand, followed by dilated convolution, a Transformer encoder is used to extract the global relationship in the spliced features and the long-term dependency relationship between elements in the sequence (*Khan et al., 2022*). As shown in Fig. 5, based on the Transformer encoder, which consists of input embedding, multi-head

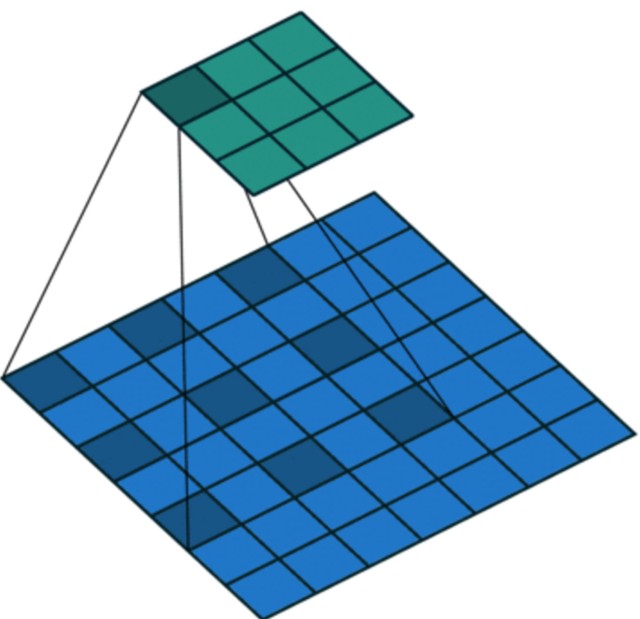

**Figure 4 Dilated convolution.**               

attention, Add & Norm, and feed forward, we incorporate positional encoding into the module. Positional encoding is an important property for sequential signals; taking the 39 DNA fragments shown in Fig. 2 as an example, if their positions or arrangement orders are changed, they will form a new DNA sequence that is totally different. Therefore, with positional encoding, word orders can be introduced to distinguish between DNA sequences.

Another important mechanism in the Transformer encoder is the multi-head attention (MHA), which computes the relative importance between different positions in the input sequence so as to provide better input feature representation for the subsequent feed forward network. Figure 6 depicts the framework of MHA. The input features, which are in the form of 3D tensors, are copied multiple times. For each feature, a weighting factor is calculated with the self-attention mechanism, with which the weighted summation of the input features are calculated. MHA can map the input features to multiple sub-spaces, and improves the model's understanding of the input sequence with feature extraction, attention calculation, and feature concatenation. Furthermore, each head in MHA works independently, thus expanding the decision space of the model and enabling better decisions while mitigating over-fitting.

Finally, after the operation of 'addition and normalization (Add & Norm)', the Transformer produces the features of the gene, which are used for classification.

## Classification

As shown in Fig. 2, the features extracted by the encoder are finally sent to the classifier to predict the sites of methylation. The classifier has three fully connected layers with dropout. The dimensions of the three fully connected layers are 128, 64, and 32,

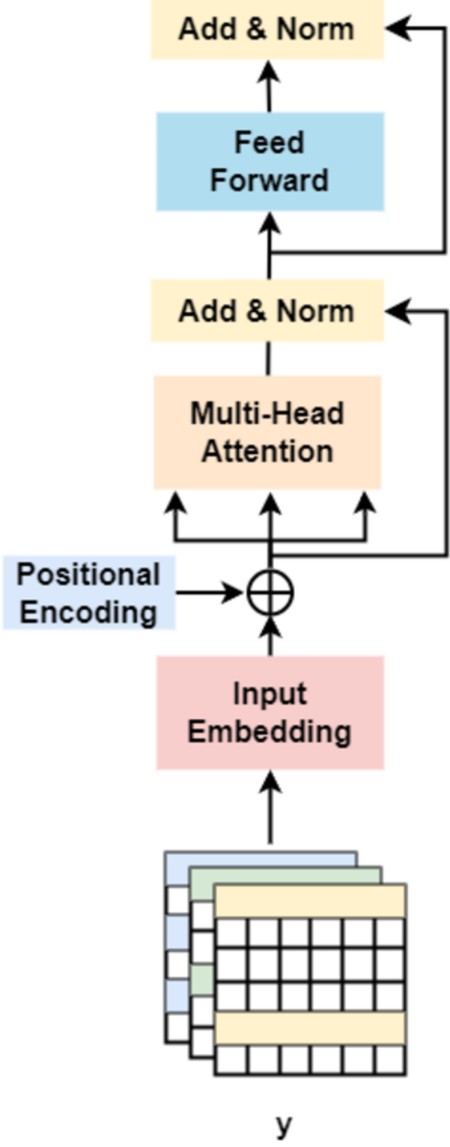

**Figure 5 Transformer encoder.**

respectively. Finally, a Sigmoid activation function is used to report whether a site is 5mc or non-5mc. In addition, the categorical cross-entropy loss is adopted to train the network.

## RESULTS AND ANALYSIS

### Dataset

In learning-based methods, the dataset is of fundamental importance. In this study, we use the Cancer Cell Line Encyclopedia (CCLE) dataset proposed by *Zhang, Xiao & Xu (2020b)*, where 5mC modification sites of various cancer cell lines are processed by a simplified RRBS experiment. Especially, we focus on investigating the distribution of 5mC sites in small cell lung cancer (SCLC) (*Barretina et al., 2012*; *Li et al., 2019*). DNA fragments with 'C' located in the center are extracted and notated as
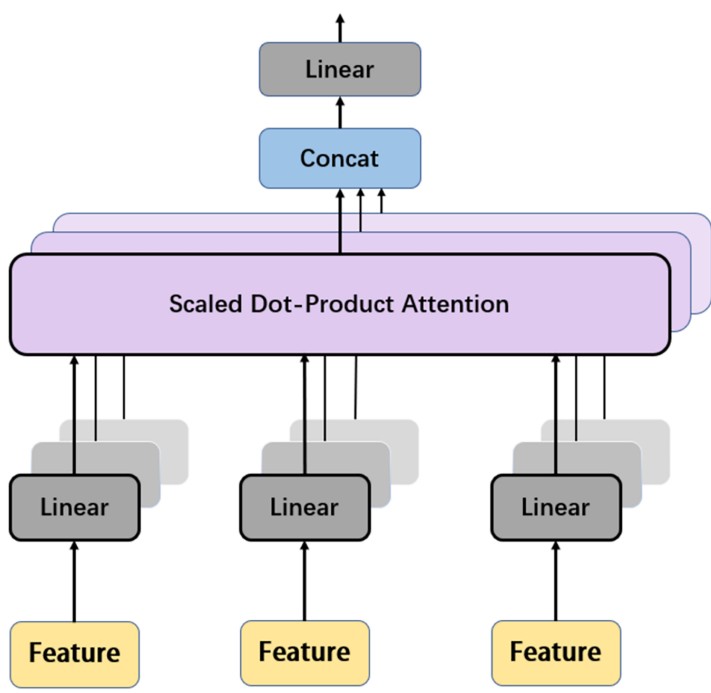

3D tensor  (batch_size, senquence_length, embedding_dim)

**Figure 6  Multi-head attention.**               

**Table 1  Details of the dataset.**

| Dataset | Positive sample | Negative sample |
|---|---|---|
| Training dataset | 55,800 | 658,861 |
| Testing dataset | 13,950 | 164,715 |
| Total | 69,750 | 823,576 |

$$E_{(\delta)}\ (C) =\ E_{-(\delta)}\ E_{-(\delta-1)}\ \cdots\ E_{-(1)}\ C\ E_{+(1)}\ \cdots\ E_{+(\delta-1)}\ E_{+(\delta)} \tag{3}$$

where for each site $E \in \{A, T, G, C\}$. In the implementation, by following the rule of WGBS, $\delta$ is set to 20, and each fragment has 41 sites. In this way, a total of 893,326 DNA fragments are obtained, including 69,750 methylation-positive samples and 823,576 negative ones. As shown in Table 1, the ratio between the negative and positive samples is about 13.3, which coincides with the distribution of 5mC in real cases.

The experiment is conducted on a server with an Intel(R) Core(TM) i9-10900F CPU, 64 GB RAM, and an NVIDIA GeForce RTX 3090 GPU. The software is programmed with Python 3.7, Keras-nightly 2.8, and tf-nightly-gpu 2.8.0.

## Performance evaluation

The model is trained and tested with the aforementioned dataset. According to the test results, *e.g.*, the numbers of true negative (TN), false negative (FN), true positive (TP), and

**Table 2 Summary of existing tools for 5mC sites prediction in genome-wide DNA promoters.**

| Method | Encoding | Feature extraction and classification |
|---|---|---|
| iPromoter-5mC | One-hot | Deep neural network |
| 5mC-Pred | K-mers | XGBoost |
| BiLSTM-5mC | One-hot and NPF | BiLSTM |
| Our model | GloVe | Digital convolution and Transformer encoder |

false positive (FP) samples, the following indexes are computed to evaluate the performance of the model.

- Sensitivity (Sen) refers to the ratio of correctly predicted positive samples to all positive samples.

$$Sen = \frac{TP}{TP + FN} \qquad (4)$$

- Specificity (Spe) refers to the ratio of correctly predicted negative samples to all negative samples.

$$Spe = \frac{TN}{TN + FP} \qquad (5)$$

- Accuracy (Acc) refers to the ratio of correctly classified samples, both positive and negative, to all tested samples.

$$Acc = \frac{TP + TN}{TP + TN + FP + FN} \qquad (6)$$

- The Matthews Correlation Coefficient (Mcc) considers the joint relationship between TP, TN, FP, and FN, and comprehensively evaluates the consistency between the predicted results and the ground truth.

$$Mcc = \frac{TP \times (TN) - FP \times (FN)}{\sqrt{(TP + FP) \times (TP + FN) \times (TN + FP) \times (TN + FN)}} \qquad (7)$$

- Area under the curve (AUC) compares the performance of different models by calculating the area under the Receiver Operating Characteristic (ROC) curve, and a larger value indicates a higher degree of authenticity.

## Performance comparison with SOTA methods

Three state-of-the-art (SOTA) methylation prediction methods, iPromoter-5mC (*Cheng et al., 2021*), 5mC-Pred (*Nguyen et al., 2021*) and BiLSTM-5mC (*Zhang, Xiao & Xu, 2020b*), are compared with our model. Table 2 presents the technique features, including encoding, feature extraction, and classification of the methods. The aforementioned dataset proposed by *Zhang, Xiao & Xu (2020b)* is used to evaluate the performance of the models, and the results are shown in Fig. 7. [1]

[1] Note that to present the best performance indexes, the results of SOTA methods are directly referenced as in *Zhang, Xiao & Xu (2020b)*, while the proposed method is trained with the same dataset and configuration.

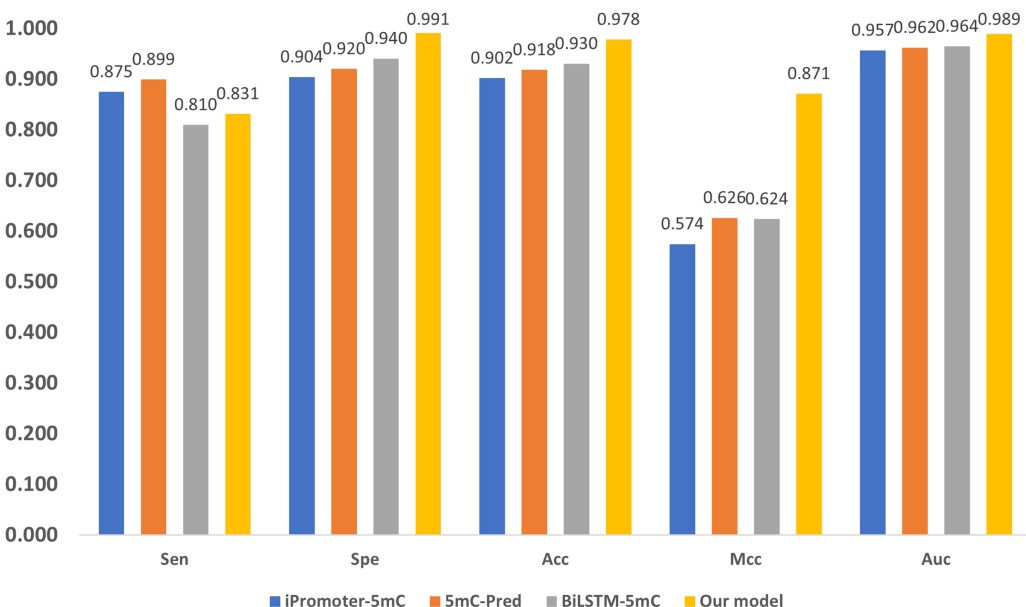

**Figure 7 Performance comparison with SOTA methods.**

As shown in Fig. 7, our model performs the best in terms of Spe, Acc, Mcc, and Auc, indicating that our model can get more essential features and the classifier is also more accurate. Our model adopts encoding techniques, including word embedding and GloVe, and Transformer feature extraction, as well as dilated convolution. These techniques improve the ability in modeling the relationship among sub-sequences, which also benefits the accurate classification of methylation for the gene sites.

We also notice that, in terms of 'Sen', the proposed framework is slightly lower than iPromoter-5mC and 5mC-Pred, the reason is that our method focuses on making reliable predictions, or in other words our model is tend to classify a positive sample as negative if it is not that confident. As a result, 'TP' becomes slightly smaller, and 'FN' is larger than the ground truth. Although this reduces the value of 'Sen', it makes 'TP' more reliable. The high MCC index indicates that our model can more accurately classify samples, and it also shows that our model can handle uncertainties better. To be specific, when our model is uncertain about the classification of a sample, it tends to classify the sample as negative to avoid misclassification. This approach reduces false judgments, thus improving the MCC index. On the other hand, it should be noticed that, in terms of the overall accuracy 'Acc' that takes all tested samples involved, the proposed method reaches 0.978, which is about 5% higher than the sub-optimal method BiLSTM-5mC.

## Influence of encoding methods

Encoding methods have a significant impact on the model's performance. In DNA coding, GloVe encoding (*JeffreyPennington & Manning, 2014*) and one-hot encoding (*Vinyals et al., 2016*) are the most widely accepted methods, and adopted in related research (*Angermueller et al., 2017*; *Tian et al., 2019*). To compare their performance, we replace

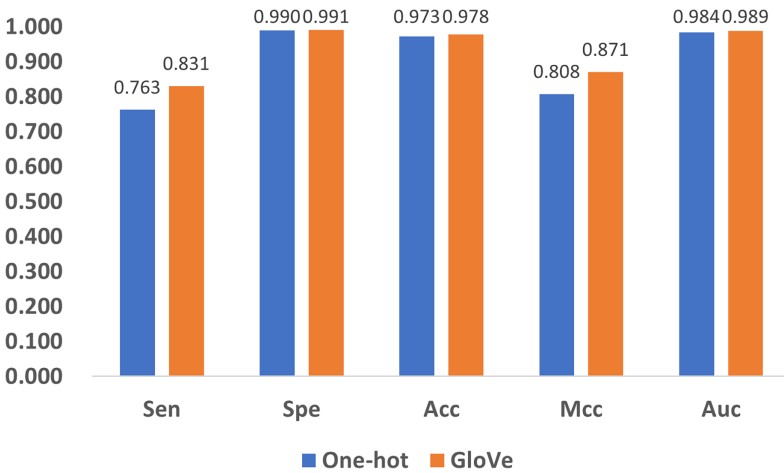

**Figure 8 Performance comparison of feature encoding methods for the prediction of 5mC sites.**

GloVe encoding in Fig. 1 with one-hot encoding, and compare the performance with the five quality indexes as shown in Fig. 8. It can be noticed that both methods produce satisfactory results, but GloVe encoding still performs better than one-hot encoding.

To be specific, for Spe, Acc, and Auc, the performance of the methods are similar with index values above 0.97. For the other two indexes, Sen and Mcc, GloVe encoding achieves significant performance improvement over one-hot encoding. The reason is that one-hot encoding only provides the simplest mapping of the four bases A, T, C, and G, resulting in a low-dimensional representation of DNA, while GloVe encoding incorporates sliced DNA fragments, and thus better represents the relationship among sub-sequences. Therefore, GloVe encoding exhibits a better ability to identify positive examples, as well as a higher correlation between the predictions and the ground truth.

## Influence of feature extraction methods

In addition to encoding, feature extraction methods also greatly affect the methylation detection results. The long short term memory (LSTM) (*Yu et al., 2019*) and gated recurrent unit (GRU) (*Dey & Salem, 2017*) are the most widely used methods in extracting features for 1D sequences, and thus we compare LSTM, GRU and the proposed Transformer encoder. As shown in Fig. 9, it can be noticed that the Transformer encoder outperforms LSTM and GRU in terms of Sen, Mcc, and Auc. The reason is that, as a recurrent neural network, LSTM relies on memory units to transmit information when dealing with long sequences. However, as the sequence grows longer, the information transmission becomes weaker in the network, which impairs the ability of long-term modeling. Compared to LSTM, GRU loses one gate unit, and is less suitable for extracting long-term dependencies, which leads to poorer performance in terms of Sen and Mcc. However, with a combination of update and reset gates, it dynamically adjusts the importance of the historical data in the hidden state, thereby improving the performance. As a result, it also achieves similar metrics to LSTM, such as Spe, Acc, and Auc. In comparison, the Transformer encoder utilizes the multi-head attention mechanism,
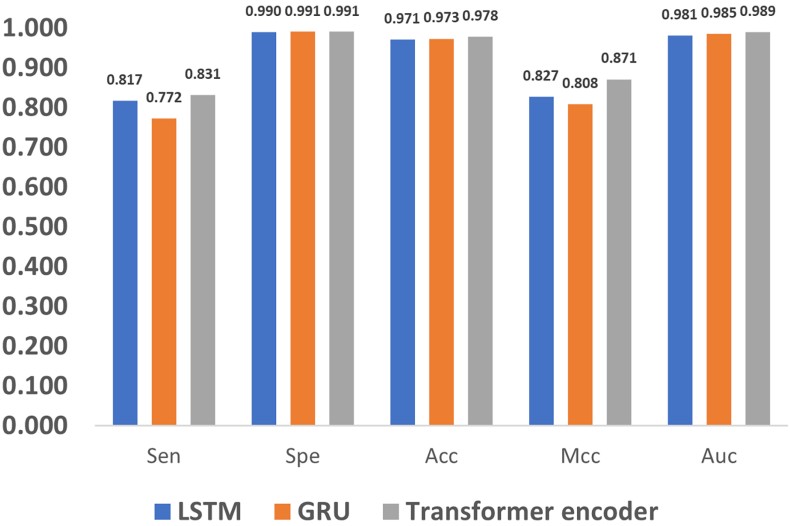

**Figure 9 Performance evaluation of different feature extraction methods.**

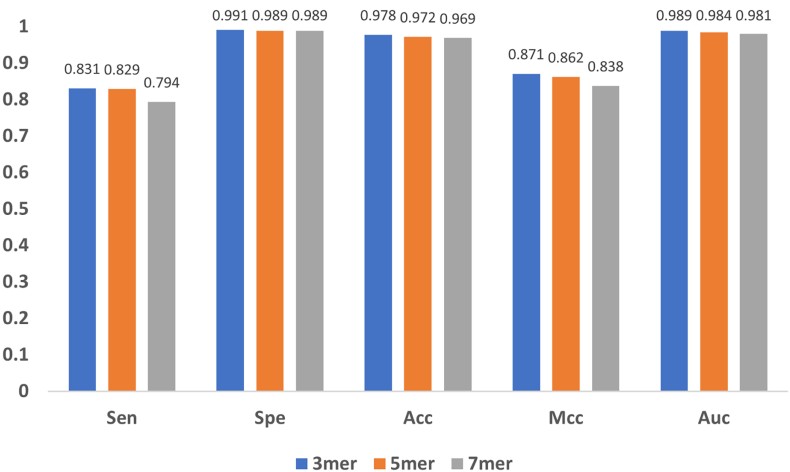

**Figure 10 Influence of sub-sequence length _k_.**

which directly models the relationship between input signals without relying on the context, and thus better deals with long sequence data.

## Influence of sub-sequence length and GloVe characteristic length

In addition to the encoding method and the feature extraction module, the length of the sub-sequence, _e.g._, the value of $k$ in $k-$mers, and the characteristic length of GloVe also affect the results. As $k$ increases, the number of sub-sequences increases exponentially, leading to great even un-acceptable computational burden, and thus we only compared the cases when $k$ equals to 3, 5 and 7. As shown in Fig. 10, the results indicate that the model performs stable with minor changes in $k$, suggesting that the model is not sensitive to $k$. Note that, $k \geq 3$ must be satisfied, because for every base, the contextual relationship
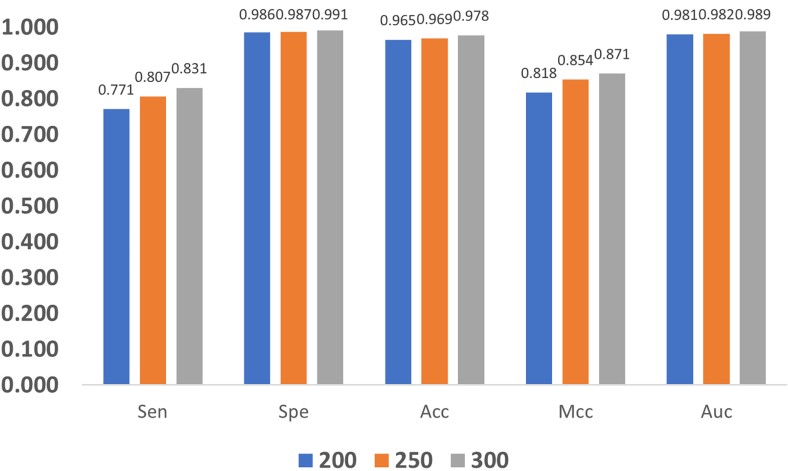

**Figure 11 Influence of GloVe characteristic length.**

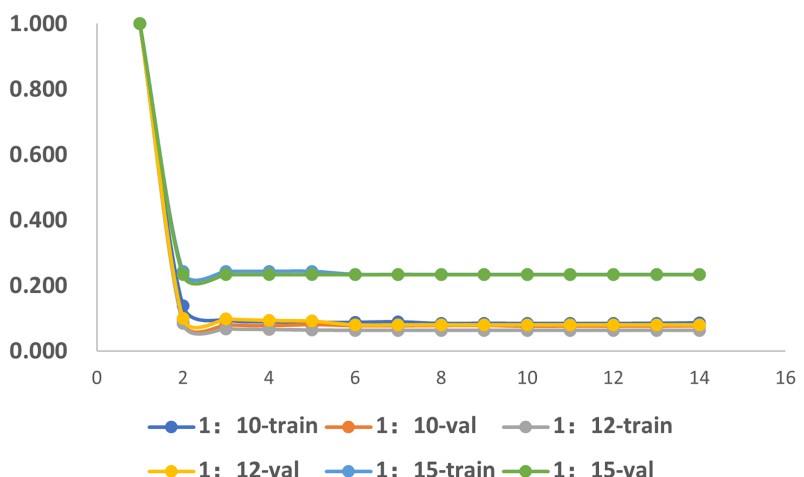

**Figure 12 The training progress of datasets with different ratios.** Train, training set; val, validation set.

between it and its preceding/following bases must be established, and at least three bases are needed.

In addition, we also compare the impact of GloVe characteristic length on the model. As shown in Fig. 11, as the length increases, the quality metrics also increase, and this indicates that larger length values are more desired. Nevertheless, increasing the length also brings a heavier computation burden, and finally, the GloVe characteristic length is empirically set to 300 by balancing performance and complexity.

## The impact of imbalance of dataset

To investigate the influence of dataset imbalance, we test three datasets with positive-to-negative ratios of 1:10, 1:12, and 1:15, respectively, and the ratio of 1:12 is the most widely adopted one in benchmark datasets. Figure 12 shows that our model reaches fast convergence in all datasets, and Fig. 13 proves that our model exhibits good stability and

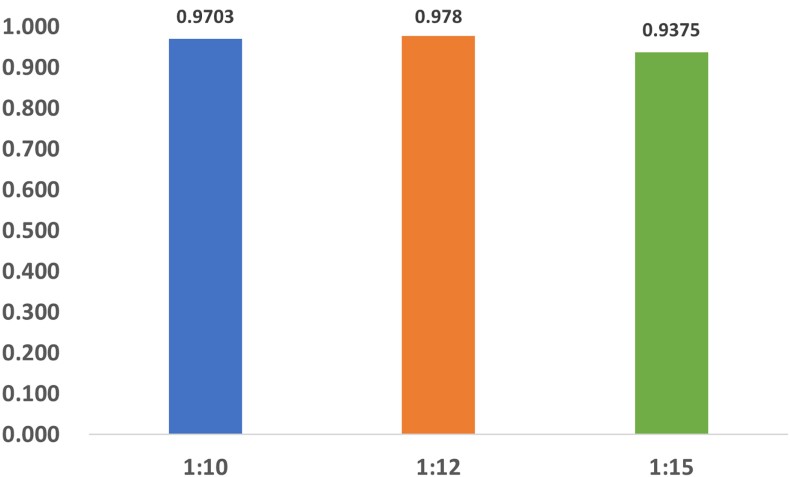

**Figure 13 The accuracy of the datasets with different ratios.**

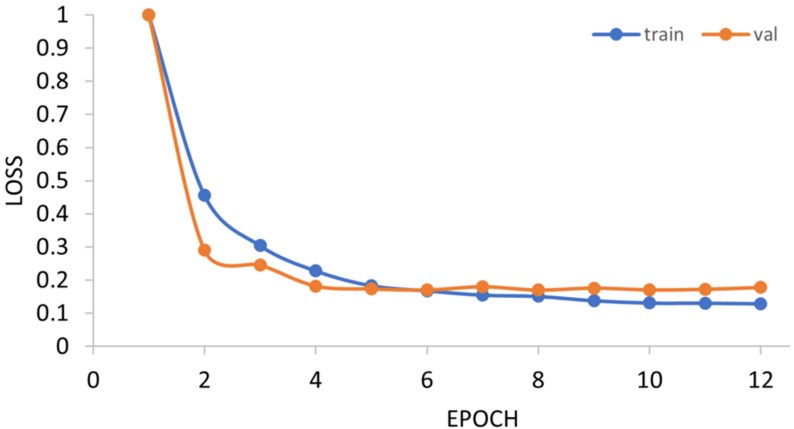

**Figure 14 Training progress of the m1A dataset.** Train, training set; val, validation set.

robustness against data imbalance. Compared with the widely accepted benchmark of 1:12, the accuracy drop is less than 0.01 for the 1:10 dataset and less than 0.05 for the 1:15 dataset.

## Generality on m1A data

The generalization ability of the proposed method is also tested by extending the model to applications of other types of DNA data. For example, in Figs. 14 and 15, the proposed model is tested with m1A data of the EMDLP dataset (*Wang et al., 2022*). It can be noticed that the network converges after 12 epochs, and the accuracy reaches 95.8%. This demonstrates that the proposed model has good generation performance and is promising to be applied to different DNA data.

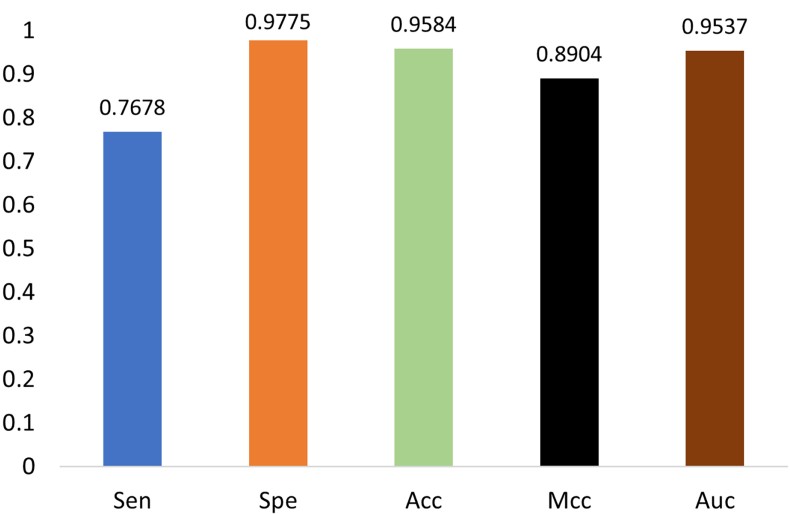

**Figure 15 Performance on m1A dataset.**

## CONCLUSION

Methylation detection for DNA sequences is an important task in epigenetics. It also serves as a cancer diagnostic biomarker, a therapeutic target, and a prognostic evaluation criterion. Traditional methylation detection methods are based on wet experiments, which are time and financial consuming. In recent years, deep learning has facilitated methylation detection in a signal processing manner. However, the state-of-the-art methods still face challenges in improving the accuracy and the robustness.

In this article, we propose DeepMethylation, a novel methylation prediction approach based on deep learning. On the one hand, we propose a new DNA representation format with word embedding and GloVe, which improves the ability in modeling the relationship between DNA sub-sequences. On the other hand, we introduce dilated convolution and Transform encoder to better extract both local and global features, especially in dealing with the relationship between remote DNA sequences.

Experimental results have demonstrated that the proposed method reaches an accuracy of 97.8%, which greatly outperforms the SOTA methods. In addition to the accuracy, the influence of encoding methods, feature extraction methods, sub-sequence length, GloVe characteristic length, and imbalance of dataset on the methylation prediction results are also studied. The results proved that GloVe encoding exhibits a better ability to identify positive samples and has a higher correlation between the predictions and the ground truth. Transformer encoder utilizes the multi-head attention mechanism to directly model the relationship between input signals without relying on context and achieves higher accuracy than LSTM and GRU. As the length parameters, optimal performance is achieved when the sub-sequence length is set to 3 and GloVe feature length is 300. Last but not least, we also investigated the robustness with different ratios of positive and negative samples, and our model exhibits good robustness.

To the best of our knowledge, this is the first work that applies Transformer to 5mc methylation prediction, and the results are promising in accuracy, which demonstrates that deep learning models can be further explored in gene sequence research and other related tasks. In the future, further investigation on larger datasets and other types of biology data is still necessary. On the other hand, novel theoretical and experimental supports are still to be developed, such as developing unsupervised learning methylation detection approaches which do not need ground truth labels with expensive and labor-intensive wet experiments.

### Funding

This work was supported by the Natural Science Foundation of Hubei Province (No. 2022CFB349). The funders had no role in study design, data collection and analysis, decision to publish, or preparation of the manuscript.

### Grant Disclosures

The following grant information was disclosed by the authors:
Natural Science Foundation of Hubei Province: 2022CFB349.

### Competing Interests

The authors declare that they have no competing interests.

### Author Contributions

- Zhe Wang conceived and designed the experiments, performed the experiments, analyzed the data, prepared figures and/or tables, authored or reviewed drafts of the article, and approved the final draft.
- Sen Xiang conceived and designed the experiments, analyzed the data, prepared figures and/or tables, authored or reviewed drafts of the article, and approved the final draft.
- Chao Zhou conceived and designed the experiments, analyzed the data, authored or reviewed drafts of the article, and approved the final draft.
- Qing Xu analyzed the data, authored or reviewed drafts of the article, and approved the final draft.

### Data Availability

The data is available at NCBI, GitHub and Zenodo:
- PRJNA523380
- https://github.com/sb111169/tf-5mc/tree/main/shuju.
- Zhe Wang. (2023). DeepMethylation. Zenodo. https://doi.org/10.5281/zenodo.8191512.
The code is available in the Supplemental File and at GitHub:
- https://github.com/sb111169/tf-5mc/tree/main.

## Supplemental Information

Supplemental information for this article can be found online at http://dx.doi.org/10.7717/peerj.16125#supplemental-information.

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
