# Peer review of "DeepMethylation: a deep learning based framework with GloVe and Transformer encoder for DNA methylation prediction"

_PeerJ, doi:10.7717/peerj.16125_

## Round 0.1 · original submission · Major Revisions

Dear authors, please refer to the reviewers' comments and revise your manuscript accordingly. When resubmitting your manuscript. please provide a point-by-point response letter so the reviewers and editor can see your responses to their comments.

Reviewer 1 ·

Basic reporting

This study proposes a deep learning prediction method for DNA methylation. DNA sequence encoding and feature extraction are performed by combining GloVe and transformer encode. In the 5mC dataset, a prediction accuracy of 97.9% was achieved, and the performance was better than that of previous studies, and the feasibility of the modified method was verified in the m1A dataset. This research is relatively complete, but the manuscript needs to be further polished before it could be considered for publication by the journal.

Line 79-82, "convolutional neural networks (CNNs) can extract features for DNA, but they are not sensitive to 1D sequential data. On the other hand, recurrent neural networks (RNNs) are more suitable for feature extraction of sequential signals, but they do not perform well in learning81 remote relationship", the disadvantages of CNN and RNN are described, but in Line 131-133, "CNN and recurrent neural networks RNN have been proven to perform well in predicting DNA methylation sites" are also described. This is contradictory, author should rationally explain the advantages and disadvantages of these methods.

Experimental design

no comment

Validity of the findings

Line 251-252, "In order to make a fair comparison, all models are trained with the aforementioned dataset", I checked the paper, and found that the data provided in Table 1 is different from the total amount of data mentioned in these papers. Whether the author used data quality control or pre-processing methods to screen the data (10.3390/molecules26247414) (10.3389/fcell.2020.00614)?
Line 251-252, if the amount of data used is different from the three references cited, "iPromoter-5mC (Cheng et al., 2021), 5mC-Pred (Tran et al., 2021) and BiLSTM-5mC (Zhang et al., 2020b)", then whether the conclusion of Figure 8-12 is reasonable. In addition, I'm curious whether the author trained the three methods mentioned in Table 2, because I saw the same table and results in the references (10.3390/molecules26247414), if the result is a reference, it should be clearly stated in the paper to avoid unnecessary misunderstanding.
In the two subsections "Influence of encoding methods" and "Influence of feature extraction methods", by comparing with one-hot encoding and LSTM, what is the reason why the author chooses to compare with these two methods and whether there are other methods worth comparing? I think it is not enough to compare only one method.
The method proposed in this study has achieved excellent results, and its performance exceeds that of similar studies. The MCC index score is far higher than that of the other three methods. The author should discuss the reasons for the high performance of this research model in detail.

Annotated reviews are not available for download in order to protect the identity of reviewers who chose to remain anonymous.

·

Basic reporting

Line 262 - check for proper usage of the word 'trend' otherwise basic reporting looks appropriate.

Experimental design

Experimental design is appropriate. Great work of comparing the different model on the same dataset as the proposed model.

Validity of the findings

Finding and conclusions look appropriate.

Additional comments

Small Cell Lung Cancer is a great start and would be curious to see its application to other tumor types as well.

Reviewer 3 ·

Basic reporting

Figures 8 to 12 can be merged together and labelled appropriately.

Experimental design

1. It is recommended to add some ablation experiments, for example, when K in k-mer is set to 2 or 4, the glove Characteristic length is 50 or 150
2. It is recommended to discuss whether the imbalance of the dataset has an impact on the model results;
3. Figure 13 discusses the comparison between one hot encoding and the encoding method proposed in the article. Figure 8 shows the results obtained using one hot and deep neural networks. Do these two results indicate that the transformer method proposed in this article is not as good as deep neural networks? Please explain this.

Validity of the findings

The conclusion section is insufficient, and it is recommended to conduct in-depth discussions on the results of the article

---

## Round 0.2 · accepted · Accept

All the reviewers have endorsed the publication of the study.

Reviewer 1 ·

Basic reporting

no comment

Experimental design

no comment

Validity of the findings

no comment

·

Basic reporting

Basic reporting is appropriate

Experimental design

Methods are well written and is appropriate.

Validity of the findings

A thorough analysis of the lung cancer data with performance comparison helps validate how the transformer encoder plus dilated convolution better extracts features with respect to DNA sequences.

Reviewer 3 ·

Basic reporting

no comment

Experimental design

no comment

Validity of the findings

no comment

Additional comments

It is expected that the authors will explore the feature representation of other NLP models in the bioinformatics domain in the future.

---

## Author Rebuttal · Round 0.2

# Response to Reviewers' Comments

We thank the editor and the reviewers for their insightful comments, and we have revised the manuscript based on their suggestions. Those comments are all valuable and very helpful for revising and improving our manuscript, as well as providing important guiding significance to our research. We have studied the comments carefully and have made the corrections, which we hope will meet with approval. All revisions are marked in red in the new manuscript. Detailed replies to their comments and how we have taken care of these points in the revised manuscript are given below.

## Responses to Editor's comments

**[Comment 1]:** You have designated equal co-authors in the submission system, but not on the manuscript document. If your manuscript is accepted, we will only use the information entered in the system. Please check the submission system to ensure that the correct authors have been selected.

**[Response]:** Thank you for your attention and feedback. We will ensure to update the manuscript document to reflect the designated equal co-first authorship information entered in the submission system.

**[Comment 2]:** Thank you for providing a link to your code repository (https://github.com/sb111169/tf-5mc/tree/main). If you created the code yourself, then in accordance with our open data policy, we require a linked DOI to this repository. This can be generated through the data archiving tool Zenodo.

**[Response]:** Thank you for your attention and feedback. We have generated the corresponding DOI and filled it into the system.

**[Comment 3]:** We note you've manually tracked your changes. Please could you upload the manuscript with computer-generated tracked changes to the Revision Response Files section. The reviewers and Academic Editor will want to see all of the changes documented and will normally request it if some changes appear to be missing. Please use latexdiff to show changes between latex documents https://www.overleaf.com/learn/latex/Articles/Using_Latexdiff_For_Marking_Changes_To_Tex_Documents.

**[Response]:** Thank you for your attention and feedback. We have generated all the change record files and uploaded them to the system.

**[Comment 4]:** In the reference section, please provide the full author name lists for any references with 'et al.' including, but not limited to, these references.

**[Response]:** Thank you for your attention and feedback. We have completed the author information.

**[Comment 5]:** Please reformat the y-axes of your bar graphs in Figures 9 so that the axes start at zero. Beginning the y-axis of a bar graph at a non-zero value can be misleading as bar graphs compare absolute values. Please do not add breaks in the y-axes to correct this issue.

**[Response]:** Thank you for your attention and feedback. We have redrawn Figure 9 and updated it in the tex file.

**[Comment 6]:** The figures must be uploaded in the Primary Files section, or be contained in the .tex file, not included in the zip file for the LaTeX source files (only the LaTeX source files (.tex, .bib, .cls, etc.) can be contained in a zip file folder). Please remove the figure files from the zip file, and upload them in the **Primary Files section**. The Figures must be named using the Figure numbers.

**[Response]:** Thank you for your attention and feedback. We have deleted the images from the zip file and uploaded them to the main file section.

**[Comment 7]:** Please use numbers to name your files, example: Fig1.eps, Fig2.png.

**[Response]:** Thank you for your attention and feedback. We have renamed the images.

# Responses to Reviewer #1's comments

**[Comment 1 (General Comment)]:** This study proposes a deep learning prediction method for DNA methylation. DNA sequence encoding and feature extraction are performed by combining GloVe and transformer encoder. In the 5mC dataset, a prediction accuracy of 97.9% was achieved, and the performance was better than that of previous studies, and the feasibility of the modified method was verified in the m1A dataset. This research is relatively complete, but the manuscript needs to be further polished before it could be considered for publication by the journal. The following points should be addressed point-by-point in details.

**[Response]:** We appreciate the helpful comments of the reviewer. The suggestions from the reviewer help us to improve our manuscript significantly. According to the reviewer's comments, we have tried our best to revise the manuscript. We hope our revised manuscript can meet the criteria of acceptance.

**[Comment 2]: Line 79-82**, "convolutional neural networks (CNNs) can extract features for DNA, but they are not sensitive to 1D sequential data. On the other hand, recurrent neural networks (RNNs) are more suitable for feature extraction of sequential signals, but they do not perform well in learning remote relationship", the disadvantages of CNN and RNN are described, but in **Line 131-133**, "CNN and recurrent neural networks RNN have been proven to perform well in predicting DNA methylation sites" are also described. This is contradictory, author should rationally explain the advantages and disadvantages of these methods.

**[Response]:** Thanks for the comment. We are sorry that the language is inaccurate that leads to misunderstanding and makes the reviewer confused. The performance of deep learning methylation detection methods can be evaluated in two aspects. (1) On the one hand, deep learning methods such as CNN and RNN do have shown significant improvements compared to traditional machine learning approaches. (2) On the other hand, due to their inherent characteristics, deep learning models including CNN and RNN are more focused on capturing local features, which leads to certain limitations in learning long-range feature relationships of DNA, which still need further improvement and is also the motivation of our work.

In the revised version, we have improved the language and emphasizing that CNN and RNN networks perform better than traditional machine learning methods, but still needs to be improved in long-term modeling.

**[Revision]:** (1) **Line 77-79:** On the one hand, convolutional neural networks (CNNs) are not sensitive to 1D sequential data such as DNA. On the other hand, recurrent neural networks (RNNs) perform better in extracting sequential features, but they are not good at exploring relationships for bases far away.

(2) **Line 126-130**, Different from traditional machine learning that highly relies on the experience of the researchers, deep learning methods, including convolutional neural networks (CNN) and recurrent neural networks (RNN), can automatically learn the essential features from the raw sequential data, and construct end-to-end models, which have been proven to outperform traditional machine learning methods in predicting DNA methylation sites.

**[Comment 3]: Line 251-252**, "In order to make a fair comparison, all models are trained with the aforementioned dataset", I checked the paper, and found that the data provided in **Table 1** is different from the total amount of data mentioned in these papers. Whether the author used data quality control or pre-processing methods to screen the data (10.3390/molecules26247414) (10.3389/fcell.2020. 00614)?

**[Response]:** Thanks for the comment. We are sorry for the flaws in the experiment. When processing the data, only the positive and negative samples were taken into account, while the subtle differences in the total number of samples were ignored. To correct the mistake, we re-conduct the experiment, and the proposed method has a performance gain of 4.8% over the second best method. We really appreciate the reviewer point this error out and help us make the paper more religious.

**[Revision]:** We updated the dataset and experimental results.

[Figure]

Table 1. Details of the dataset.

| Dataset | Positive Sample | Negative Sample |
|---|---|---|
| Training Dataset | 55800 | 658861 |
| Testing Dataset | 13950 | 164715 |
| Total | 69750 | 823576 |

**Figure 7.** Performance comparison with SOTA methods.

**[Comment 4]: Line 251-252**, if the amount of data used is different from the three references cited, "iPromoter-5mC (Cheng et al., 2021), 5mC-Pred (Tran et al., 2021) and BiLSTM- 5mC (Zhang et al., 2020b)", then whether the conclusion of **Figure 8-12** is reasonable. In addition, I'm curious whether the author trained the three methods mentioned in **Table 2**, because I saw the same table and results in the references (10.3390/molecules26247414), if the result is a reference, it should be clearly stated in the paper to avoid unnecessary misunderstanding.

**[Response]:** Thanks for the comment. (1) According to the suggestion of the reviewer, we have re-conduct the experiment, where the data is exactly the same with the three references cited, "iPromoter-5mC (Cheng et al., 2021), 5mC-Pred (Tran et al., 2021) and BiLSTM- 5mC (Zhang et al., 2020b)". Compared with the results of the previous version, the new results with very minor decrease, and the proposed method still has 4.8% gain over the second best method. We are sorry for the flaws in the experiment, and really appreciate the reviewer point this out. (2) We have made a solemn declaration in the article regarding the utilization of experimental results incorporating BiLSTM- 5mC.

**[Revision]:** (1) We updated the dataset and experimental results.

[Figure]

Table 1. Details of the dataset.

| Dataset | Positive Sample | Negative Sample |
|---|---|---|
| Training Dataset | 55800 | 658861 |
| Testing Dataset | 13950 | 164715 |
| Total | 69750 | 823576 |

Figure 7. Performance comparison with SOTA methods.

(2)**Footnote 1 on page 8**, Note that to present the best performance index, the results of SOTA methods are directly referenced as in (Zhang et al., 2020b), while the proposed method is trained with the same dataset and configuration.

**[Comment 5]:** In the two subsections "Influence of encoding methods" and "Influence of feature extraction methods", by comparing with one-hot encoding and LSTM, what is the reason why the author chooses to compare with these two methods and whether there are other methods worth comparing? I think it is not enough to compare only one method.

**[Response]:** Thanks for the comment. Encoding methods and feature extraction methods are fundamental issues in machine learning.

(1)As to encoding methods, DNA, as a one-dimensional sequence, currently the most widely used and mainstream encoding approaches are one-hot and GloVe encoding. These two encoding methods are also applied in other related works such as DeepCpG (Angermueller et al., 2017) and MRCNN (Tian et al., 2019). Therefore, in this paper we compare the performance of GloVe encoding and one-hot encoding.

(2)As to feature extraction methods, LSTM, GRU and Transformer are the most commonly used and widely recognized in the academic community. These methods have a certain degree of flexibility, and the choice of parameters can also affect their performance. Therefore, compared to the original manuscript, we have added a comparison with GRU. Furthermore, we have further analyzed the impact of different values of k in k-mer , GloVe characteristic length and the imbalance of the dataset on the model.

**[Revision]: (1) Line 269-273.** In DNA coding, GloVe encoding and one-hot encoding are the most widely accepted methods, and adopted in related research (Angermueller et al., 2017)(Tian et al., 272 2019). To compare their performance, we replace GloVe encoding in figure1 with one-hot encoding, and compare the performance with the five quality indexes as shown in figure 8.

[Figure]

**Figure 8.** Performance comparison of feature encoding methods for the prediction of 5mC sites.

(2) **Line 284-287**, The long short term memory (LSTM) and gated recurrent unit (GRU) are the most widely used methods in extracting features for 1D sequences, and thus we compare LSTM, GRU and the proposed Transformer encoder. As shown in figure9, it can be noticed that the Transformer encoder outperforms LSTM and GRU in terms of Sen, Mcc, and Auc.

[Figure]

**Figure 9.** Performance evaluation of different feature extraction methods.

(3)**Line 290-294**, Compared to LSTM, GRU loses one gate unit, and is less suitable for extracting long-term dependencies, which leads to poorer performance in terms of Sen and Mcc. However, with a combination of update and reset gates, it dynamically adjusts the importance of the historical data in the hidden state, thereby improving the performance. As a result, it also achieves similar metrics to LSTM, such as Spe, Acc, and Auc.

(4)**Line 298-308**, In addition to the encoding method and the feature extraction module, the length of the sub-sequence, e.g. the value of k in k-mers, and the characteristic length of GloVe also affect the results. As k increases, the number of sub-sequences increases exponentially, leading to great even un-acceptable computational burden, and thus we only compared the cases when k equals to 3, 5 and 7. As shown in figure 10, the results indicate that the model performs stable with minor changes in k, suggesting that the model is not sensitive to k.

Note that, k >= 3 must be satisfied, because for every base, the contextual relationship between it and its preceding/following bases must be established, and at least three bases are needed.

In addition, we also compare the impact of GloVe characteristic length on the model. As shown in figure 11, as the length increases, the quality metrics also increase, and this indicates that larger length values are more desired. Nevertheless, increasing the length also brings a heavier computation burden, and finally, the GloVe characteristic length is empirically set to 300 by balancing performance and complexity.

[Figure]

[Figure]

**Figure 10.** Influence of sub-sequence length $k$.

**Figure 11.** Influence of GloVe characteristic length.

(5)**Line 310-315**, To investigate the influence of dataset imbalance, we test three datasets with positive-to-negative ratios of 1:10, 1:12, and 1:15, respectively, and the ratio of 1:12 is the most widely adopted one in benchmark datasets. Figure 12 shows that our model reaches fast convergence in all datasets, and figure 13 proves that our model exhibits good stability and robustness against data imbalance. Compared with the widely accepted benchmark of 1:12, the accuracy drop is less than 0.01 for the 1:10 dataset and less than 0.05 for the 1:15 dataset.

[Figure]

**Figure 12.** The training progress of datasets with different ratios. Train: training set, val: validation set.

**Figure 13.** The accuracy of the datasets with different ratios.

**[Comment 6]:** The method proposed in this study has achieved excellent results, and its performance exceeds that of similar studies. The MCC index score is far higher than that of the other three methods. The author should discuss the reasons for the high performance of this research model in detail.

**[Response]:** Thanks for the comment. In the revised manuscript, we have provided a detailed discussion on the specific reasons that may contribute to the high performance of our model.

**[Revision]: Line 261-264**, The high MCC index indicates that our model can more accurately classify samples, and it also shows that our model can handle uncertainties better. To be specific, when our model is uncertain about the classification of a sample, it tends to classify the sample as negative to avoid misclassification. This approach reduces false judgments, thus improving the MCC index.

# The responses to Reviewer #2's comments

**Comments from the reviewer:**

**[Comment 1]:** Line 262 - check for proper usage of the word 'trend' otherwise basic reporting looks appropriate.

**[Response]:** Thanks for the comment. We are sorry for the typo. The word 'trend' should be 'tend'. We have corrected the mistake. Thank you once again to the reviewer for pointing out this mistake.

**[Revision]: Line 259**, our model is tend to classify a positive sample as negative if it is not that confident.

**[Comment 2]:** (1) Experimental design is appropriate. Great work of comparing the different model on the same dataset as the proposed model. (2) Finding and conclusions look appropriate.

**[Response]:** (1) Thank you for your positive feedback on our experimental design. Your acknowledgment motivates us to continue our research in this direction. Thank you once again for your valuable input. (2) Thank you for your feedback on our findings and conclusions. We have dedicated considerable effort to ensure the accuracy and validity of our results. Your positive assessment validates our hard work. We are grateful for your thorough review and valuable input.

**[Comment 3]:** Small Cell Lung Cancer is a great start and would be curious to see its application to other tumor types as well.

**[Response]:** Thank you for your kind words regarding our work on Small Cell Lung Cancer. We appreciate your interest in our research and your curiosity about its potential application to other tumor types. We also believe that our approach holds promise for broader applications across different cancer types. We will take your suggestion into consideration and explore the possibilities of extending our study to other tumor types. Your feedback has provided us with valuable insights and we are grateful for your thoughtful review.

# The responses to Reviewer #3's comments

**[Comment 1]:** Figures 8 to 12 can be merged together and labelled appropriately.

**[Response]:** Thanks for the comment. We have merged the figures together to enhance the clarity and organization of our paper.

**[Revision]:** The merged figures 8 to 12, which is figure 7 in the revised manuscript.

[Figure]

**Figure 7.** Performance comparison with SOTA methods.

**[Comment 2]:** It is recommended to add some ablation experiments, for example, when k in k-mer is set to 2 or 4, the glove characteristic length is 50 or 150.

**[Response]:** Thanks for the comment.

(1) As to the value of k in k-mer, we have compared the cases where k is equal to 3, 5 and 7. Note that for every base, we need to model the relationship for between it and its prior and successive bases, and thus $k \geq 3$ must be satisfied.

(2) As to glove characteristic length, we compared the cases where the length is 200, 250 and 300. The results show that larger length can improves the performance, but the complexity also grows rapidly. According to the empirical values obtained from previous studies(JeffreyPennington and Manning, 2014), it has been proven that the model achieves the best performance when the length is 300.

**[Revision]: Line 298-308**, In addition to the encoding method and the feature extraction module, the length of the sub-sequence, e.g. the value of k in k-mers, and the characteristic length of GloVe also affect the results. As k increases, the number of sub-sequences increases exponentially, leading to great even un-acceptable computational burden, and thus we only compared the cases when k equals to 3, 5 and 7. As shown in figure 10, the results indicate that the model performs stable with minor changes in k, suggesting that the model is not sensitive to k.

Note that, $k \geq 3$ must be satisfied, because for every base, the contextual relationship between it and its preceding/following bases must be established, and at least three bases are needed.

In addition, we also compare the impact of GloVe characteristic length on the model. As shown in figure 11, as the length increases, the quality metrics also increase, and this indicates that larger length values are more desired. Nevertheless, increasing the length also brings a heavier computation burden, and finally, the GloVe characteristic length is empirically set to 300 by balancing performance and complexity.

[Figure]

**Figure 10.** Influence of sub-sequence length $k$.

**Figure 11.** Influence of GloVe characteristic length.

**[Comment 2]:** It is recommended to discuss whether the imbalance of the dataset has an impact on the model results;

**[Response]:** Thanks for the comment. In previous experiments, the positive-to-negative sample ratio was approximately 1:12. Based on this, we conducted experiments with positive-to-negative sample ratios of 1:10 and 1:15.

**[Revision]: Line 310-315**, To investigate the influence of dataset imbalance, we test three datasets with positive-to-negative ratios of 1:10, 1:12, and 1:15, respectively, and the ratio of 1:12 is the most widely adopted one in benchmark datasets. Figure 12 shows that our model reaches fast convergence in all datasets, and figure 13 proves that

our model exhibits good stability and robustness against data imbalance. Compared with the widely accepted benchmark of 1:12, the accuracy drop is less than 0.01 for the 1:10 dataset and less than 0.05 for the 1:15 dataset.

[Figure]

**Figure 12.** The training progress of datasets with different ratios. Train: training set, val: validation set.

**Figure 13.** The accuracy of the datasets with different ratios.

**[Comment 3]:** Figure 13 discusses the comparison between one hot encoding and the encoding method proposed in the article. Figure 8 shows the results obtained using one hot and deep neural networks. Do these two results indicate that the transformer method proposed in this article is not as good as deep neural networks? Please explain this.

**[Response]:** Thanks for the comment. We are sorry that that the language is inaccurate that leads to misunderstanding and makes the reviewer confused. The original Figure 8 (now Figure 7) represents the comparison with SOTA methods. Our 'Sen' indicator is indeed lower than two of these methods, because our model tend to classify a positive sample as 'negative' if it is not that confident. Although this reduces the value of 'Sen', it provides more reliable judgement for 'TP'. However, our other values are higher than those of the SOTA methods. On the other hand, the original Figure 13 (now Figure 8) demonstrates that only the encoding method is changed, and the feature extraction part still uses the Transformer module. Moreover, as shown in Figure 9, it can be observed that by only replacing the feature extraction model, our Transformer module outperforms other deep learning networks. According to the results above, we can have a general conclusion that, the transformer encoder outperforms the existing methods that based on CNN and RNN.

**[Comment 4]:** The conclusion section is insufficient, and it is recommended to conduct in-depth discussions on the results of the article.

**[Response]:** Thank you for your feedback and suggestions. We appreciate your valuable input on our article. We understand your concern regarding the conclusion section and agree that further in-depth discussions on the results would enhance the quality of the paper. We have included the additional experiments in the concluding discussion section to provide a more comprehensive analysis and interpretation of the obtained results.

**[Revision]: Line 323-349**, Methylation detection for DNA sequences is an important task in epigenetics. It also serves as a cancer diagnostic biomarker, a therapeutic target, and a prognostic evaluation criterion. Traditional methylation detection methods are based on wet experiments, which are time and financial consuming.

In recent years, deep learning has facilitated methylation detection in a signal processing manner. However, the state-of-the-art methods still face challenges in improving the accuracy and the robustness.

In this paper, we propose DeepMethylation, a novel methylation prediction approach based on deep learning. On the one hand, we propose a new DNA representation format with word embedding and GloVe, which improves the ability in modeling the relationship between DNA sub-sequences. On the other hand, we introduce dilated convolution and Transform encoder to better extract both local and global features, especially in dealing with the relationship between remote DNA sequences.

Experimental results have demonstrated that the proposed method reaches an accuracy of 97.8%, which greatly outperforms the SOTA methods. In addition to the accuracy, the influence of encoding methods, feature extraction methods, sub-sequence length, GloVe characteristic length, and imbalance of dataset on the methylation prediction results are also studied. The results proved that GloVe encoding exhibits a better ability to identify positive samples and has a higher correlation between the predictions and the ground truth. Transformer encoder utilizes the multi-head attention mechanism to directly model the relationship between input signals without relying on context and achieves higher accuracy than LSTM and GRU. As the length parameters, optimal performance is achieved when the sub-sequence length is set to 3 and GloVe feature length is 300. Last but not

least, we also investigated the robustness with different ratios of positive and negative samples, and our model exhibits good robustness.

To the best of our knowledge, this is the first work that applies Transformer to 5mc methylation prediction, and the results are promising in accuracy, which demonstrates that deep learning models can be further explored in gene sequence research and other related tasks. In the future, further investigation on larger datasets and other types of biology data is still necessary. On the other hand, novel theoretical and experimental supports are still to be developed, such as developing unsupervised learning methylation detection approaches which do not need ground truth labels with expensive and labor-intensive wet experiments.